# Equine Anthelmintic Resistance: Horse Owner and Yard Manager Perception of the Barriers Affecting Strategic Control Measures in England

**DOI:** 10.3390/vetsci9100560

**Published:** 2022-10-11

**Authors:** Faye E. McTigue, Stephen C. Mansbridge, Alison Z. Pyatt

**Affiliations:** Department of Animal Health, Behaviour and Welfare, Harper Adams University, Newport, Shropshire TF10 8NB, UK

**Keywords:** equine anthelmintic resistance, strategic helminth controls, horse owners, yard managers, stakeholder collaboration, qualitative

## Abstract

**Simple Summary:**

Equine anthelmintic resistance poses a threat to the health and welfare of horses worldwide. With no new imminent anthelmintic classes, it is vital to decrease resistance rate and preserve available classes. Traditional indiscriminate anthelmintic use is not synonymous with a strategic control ideology; however, many continue to implement outdated practices. In comparison to livestock farmers, there have been few social science studies examining the behaviours of horse owners. This study aimed to understand the perceived barriers faced by horse owners and yard managers to adopting a strategic approach to helminth control, and to explore their experiences. Semi-structured interviews permitted the exploration of the stakeholders’ beliefs, opinions, experiences, attitudes, and behaviours. Personal resources, internal influences, and external influences were identified by participants as the barriers to adoption of strategic controls. Two further themes impacting effective implementation of strategic controls emerged surrounding helminth information and source, as well as accurate dosing. Challenges associated with anthelmintic administration and bodyweight estimation suggest that inaccurate dosing is prevalent. Industry wide implementation of a single ‘best practice’ could support prescriber decision-making to encourage uptake of effective strategic helminth controls. The barriers identified in this study could help influence guidance given to horse owners through a better understanding of their beliefs and behaviours.

**Abstract:**

Equine anthelmintic resistance poses a threat to the health and welfare of horses worldwide. With no new imminent anthelmintic classes, it is vital to decrease the resistance rate and preserve available classes. Traditional indiscriminate anthelmintic use is not synonymous with a strategic control ideology; however, many continue to implement outdated practices. In comparison to livestock farmers, there have been few social science studies examining horse owner behaviours. This study aimed to understand the perceived barriers faced by horse owners and yard managers to adopting a strategic approach to helminth control, and to explore their experiences. Semi-structured interviews permitted the exploration of the stakeholders’ beliefs, opinions, experiences, attitudes, and behaviours. Personal resources, internal influences, and external influences were identified by participants as the barriers to adoption of strategic controls. Two further themes impacting effective implementation of strategic controls emerged surrounding helminth information and source, as well as accurate dosing. Challenges associated with anthelmintic administration and bodyweight estimation suggest that inaccurate dosing is prevalent. Industry-wide implementation of a single ‘best practice’ could support prescriber decision-making to encourage uptake of effective strategic helminth controls. The barriers identified in this study could help influence guidance given to horse owners through a better understanding of their beliefs and behaviours.

## 1. Introduction

Helminth infections pose a significant risk to the health and welfare of humans and animals across the world [1]. Due to their insensitivity to most anthelmintics, ability to develop anthelmintic resistance (AR), prevalence, and pathogenic potential, cyathostomins are regarded as the most significant group of equine helminths [2,3]. Though some horses will tolerate infection well and have patent infections [4], mass emergence of encysted larvae from the large intestine can cause fatal larval cyathostominosis [5]. In horses younger than two years old, *Parascaris equorum* is the helminth of concern; it can cause severe clinical disease with fatal consequences due to the risk of perforation of the small intestine [6,7,8]. In horses over one year old, *Strongylus vulgaris* is considered the most significant non-cyathostomin species due to associated colic caused by non-strangulating intestinal necrosis [9]. Before the introduction of broad-spectrum anthelmintics, this helminth posed a major risk to equine health [9]. Thus, highlighting the importance of effective anthelmintics and consequences is essential should AR jeopardise their efficacy.

Equine AR was first reported in the 1960s [10] and has become increasingly prevalent worldwide. There are three anthelmintic classes authorised for equine helminth control in the UK: benzimidazoles, macrocyclic lactones, and tetrahydropyrimidines [11]; resistance status is outlined in Table 1. In the UK, there are currently no approved methods of establishing *Anoplocephala perfoliata* resistance, though there is suspicion and anecdotal reports [12]. The cause of equine AR is multifactorial, including but not limited to helminth variables such as population dynamics and biological factors; host factors such as anthelmintic pharmacokinetics and immunity; and environmental influence such as climate change, deworming frequency, underdosing, and treatment timing [10,13]. With no new anthelmintic classes forecasted, changes to anthelmintic usage are required to slow the rate of resistance [10].

Practicably applicable methods for decreasing the rate of AR include implementation of strategic control measures to reduce reliance on anthelmintics [15]. Strategic control measures include diagnostic testing to inform anthelmintic use and choice, quarantine, pasture management, and appropriate stocking density [12]. Traditional, indiscriminate seasonal anthelmintic use is not synonymous with a strategic control ideology, yet even with updated advice and information, many horse owners continue to implement outdated helminth control practices [16]. Current methods to inform anthelmintic choice include faecal worm egg counts (FWECs), lungworm faecal sedimentation test, pinworm sellotape test, *Anoplocephala* spp. blood or saliva tests, and, more recently, the encysted cyathostomins blood test. FWEC reduction tests are considered the gold standard practical method to establish AR status on yards and should be performed annually [12]. However, despite many FWEC businesses offering reduction testing free of charge [12], they are seldom used in the equine industry [17].

Unlike the livestock industry, where guidelines have been clearly identified and agreed upon by sector stakeholders—including veterinary surgeons, suitably qualified persons (SQPs), farmers, and levy boards, such as industry led Sustainable Control of Parasites in Sheep (SCOPS) [18] and Control of Worms Sustainably (COWS) [19]—there is no equivalent for the equine sector. Within the SCOPS and COWS framework, there is focus on accurate dosing to avoid treatment failure and increasing rate of AR [13].

In a broader context, livestock farmers have been the focus of social science studies exploring behaviour. To date, horse owners have not been subject to the same attention. Qualitative research into livestock farmer approaches regarding anthelmintic awareness and practices has enhanced understanding of experiences and current practices, which can help to facilitate support and change [20,21,22].

Questionnaire-based research has identified horse owner confusion surrounding interpretation of treatment guidelines, specifically quarantine [23], and highlighted a relationship between satisfaction of knowledge and implementation of strategic controls [23,24]. Furthermore, 49% of horse owners surveyed who kept their horse at livery reported that the yard controlled and implemented a common programme for all horses, and 45% of the liveries were unsatisfied with the programme [24]. Given that it is estimated that approximately 60% of leisure horses in the UK are kept at livery yards under the direction of a yard manager or owner [25], their role in influencing horse owner behaviours and in equine AR should not be underestimated.

Exploring and understanding why outdated approaches to worm control persist is essential for facilitating behavioural change and for shaping the guidance provided to horse owners and keepers. The capability, opportunity, and motivation (COM-B) model is a recognised tool for analysing behaviour in both the human and animal sectors alike [26,27,28,29]. The principles of the COM-B model explain why change within equine helminth control programmes is not widely implemented; although knowledge targets “capability,” it does not consider the “opportunity” and “motivation” factors which are vital in successfully implementing change.

This present study aimed to understand the perceived barriers faced by UK horse owners and yard managers to adopting a strategic approach to helminth control, and to explore their experiences. 

## 2. Materials and Methods

### 2.1. Study Design

A qualitative research methodology was selected to permit the exploration of the stakeholders’ beliefs, opinions, experiences, attitudes, and behaviours [30]. Purposive sampling was utilised to permit selection of information-rich participants [31] and to best increase understanding of the actual and perceived barriers faced by the stakeholders. The sample frame consisted of two stakeholder groups, defined as horse owners and yard managers.

A semi-structured interview guide, consisting of 22 questions divided across five categories, was designed, see Appendix A. The guide was informed using the contemporaneous literature, professional knowledge and expertise of the research team, and industry and organizational data. Questions were devised to address the study aims; the guide ensured research topics were covered but still allowed for participants to expand on areas of interest [32]. Participants were encouraged to engage and freely contribute throughout the interview through open questions. Following the principles of grounded theory, adjustments were made to the interview guide following each interview that were considered necessary [33]. All interviews were conducted over the telephone for interviewee convenience and enhanced participation [34]. In some cases, face-to-face interviews are considered superior to telephone interviews [35]. However, research comparing telephone and face-to-face qualitative interviewing through comparison of the transcripts yielded no significant difference [36]. Furthermore, telephone interviews may encourage participants to relax and divulge sensitive information [35], and they provide a COVID-secure option for qualitative research techniques. 

### 2.2. Data Collection

Semi-structured telephone interviews of the two stakeholder groups (see Table 2) were conducted. The interviews were conducted by the lead researcher (F.E.M.) between April and June 2022. All participants were given a plain language statement prior to the interview which outlined the nature of the research project. Informed consent was obtained prior to data collection. All data were fully anonymised and stored securely on a protected OneDrive (Microsoft Corporation, Washington, DC, USA). The data were only accessible by the researcher and research team. All data were collected and handled in accordance with the Data Protection Act 2018 and General Data Protection Regulation.

### 2.3. Data Analysis

Interviews were recorded using RingCentral (reputable communications software utilised by service businesses) and auto-transcription software Otter.ai [37]. Transcripts were checked by the lead researcher to ensure accuracy and enhance familiarisation with the data. Each participant was allocated a unique code to further ensure confidentiality (see Table 2).

### 2.4. Thematic Analysis

Thematic analysis was used to identify common themes within the interviews and to represent meaning or a pattern within the data [34]. Qualitative research techniques provide an accessible, flexible, and valuable method for exploring behaviours and experiences [34]. Thematic analysis was performed by the lead researcher using qualitative data analysis software (NVivo v12, QSR International Pty Ltd., Doncaster, Australia). The analysis software facilitated data organisation, enabling collation and construction of themes and relationships [38]. Transcript analysis was performed in accordance with the recognised six phases of thematic analysis [34].

## 3. Results

Implementation of strategic controls and blanket deworming programmes was evenly represented across the participants, with some interviewees undertaking strategic control measures and others seeking to change their current blanket deworming approach. Of all participants currently implementing strategic control measures, FWEC was the only diagnostic tool identified and only two participants implemented FWEC reduction tests to monitor anthelmintic efficacy.

Three main themes were evident and further divided into subthemes during analysis: (1) barriers to adoption of strategic controls; (2) helminth information and anthelmintic source, and (3) accurate dosing. In order to illustrate the themes, subthemes, and categories, and to express their inter-relationships, a thematic map was created to aid exploration and understanding and illustrate understanding of the data (Figure 1). Mapping the data also helped to develop the structure and provide meaning to the themes.

### 3.1. Theme 1: Barriers to Adoption of Strategic Controls

Recurring barriers to the adoption of strategic controls were indicated by horse owners and yard managers alike, and included factors such as cost, convenience, tradition and prior experience. Themes associated with perceived barriers were sub-categorised into personal resources, internal influences, and external influences.

#### 3.1.1. Subtheme 1: Personal Resources

This subtheme comprised of three main categories (1) cost, (2) convenience, and (3) labour.

The perceived cost of strategic helminth controls in comparison to just purchasing anthelmintics was clearly expressed by many horse owners and yard managers. One horse owner felt that strategic controls were a waste of money:


*It would be so expensive to strategically worm properly. It’s going to waste more of my money.*
[HO1]


*I guess the problem is the cost of the worm count is similar to that of the cost of a wormer and if you do a worm count and you still have to worm then you’re doubling the cost.*
[YM2]

Conversely, HO2 did not consider cost a barrier: 


*Probably not cost if I really believed in it or if I was really concerned about worm resistance.*
[HO2]

Cost was also linked to convenience for the horse owner:


*I suppose I thought it might be a similar cost, so it’s easier to just stick a syringe in their mouth I guess.*
[HO3]

Convenience, or the lack thereof, was also expressed in relation to the alleged ease of application and value of service:


*Manageability, the fact that I’d have to go to multiple different people to get the right things, knowing when and what to test for, I can’t get that information easily from one place that is reliable.*
[HO1]


*The fact that, although I can’t pull the information to mind immediately, the fact that I know it’s not a comprehensive solution. Often, you still have to worm because you’ve got no idea about the different worm burdens that they might be suffering from that can’t be counted on a worm egg count.*
[HO2]

Time and labour were associated with convenience by many of the interviewees:


*I just haven’t got round to it. Because it’s a fairly new business and it’s grown quite quickly and its one of the things that should have come to the forefront but, we’ve like poo picked constantly and worming, I think that has sufficed to a level, but it it’s an area I’d like to move to.*
[YM1]

The time specifically taken to do FWECs was commented on, in addition to the time and labour associated with removing faeces from pastures.


*As a large yard with a large field, poo picking is difficult, so that’s why we harrow.*
[HO4]

#### 3.1.2. Subtheme 2: Internal Influences

This subtheme comprised three categories of recurring topics relating to the internal influences faced by participants, and were described as: (1) apathy, (2) experience/tradition, and (3) knowledge. There was a lack of interest conveyed by some horse owners.


*I don’t know why really, I just haven’t. I haven’t thought about it or looked into it.*
[HO3]

HO2 also stated apathy as the only barrier. It is worth nothing that HO3 was not aware of equine AR.

Past experience, habituation, and tradition were expressed as barriers by some.


*Habit primarily, and it being the norm.*
[HO5]


*You know, it’s how we’ve always done it. And so it’s a bit of changing the yard owner’s mindset to actually let us [perform FWECs].*
[YM3]

Furthermore, inadequate knowledge of deworming and strategic principles was cited by others.


*I’m not very good with worming.*
[HO1]

Though it was not considered a current barrier, HO4 recalled knowledge being a previous barrier:


*I didn’t have a clue what worm egg counting was, until a good few years ago when it started coming around.*
[HO4]

Others’ knowledge and understanding was also commented on.


*I think the other biggest thing is that people just don’t understand it, how to worm strategically. I think not enough people know about it effectively.*
[HO1]


*The knowledge is really not there for novice owners.*
[HO4]

Dissemination of knowledge was also discussed by some.


*I think the companies selling the faecal worm egg counting kits and things, they’re selling the products that they can sell, but they can’t do a blood test. So they aren’t marketing that oh actually there’s this vital bit of information needed here.*
[HO1]


*I’ve dealt with a few vets and you have your routine vaccinations and everything else but unless somethings wrong you don’t really get pushed to talk about worming.*
[HO4]

#### 3.1.3. Subtheme 3: External Influences

This subtheme related to factors outside the control of participants presenting as barriers, and comprised one category, livery management. From a yard manager’s perspective, the volume of horses arriving with unknown histories was considered a barrier.


*I think for us, because we have, do have quite a lot of traffic. We have young horses arriving that have come from Ireland or dealers, basically places that I don’t know their worming history and I’m pretty sure that a lot of them that come to us have never ever been wormed. So I’m a little leaned towards getting a wormer in them in the first place.*
[YM2]

From a horse owner’s perspective, management of their horse felt out of their control, particularly with regards to pasture management, and this was considered a barrier. 


*I think if you’re on a yard, for example similar to mine and you’re on full livery, your horse’s management is out of your hands. If your horse, for example like mine, over winter they don’t poo pick, so I know that’s a barrier for me because if they’re not poo picking the fields I know he’s more likely to have a high worm burden an I am more likely to be required to worm.” “how my horse is managed, i.e., on full livery, I can’t control that they don’t poo pick the fields, the shared pastures on a daily basis in winter. You know, and the fields are rotated quite frequently, as in different horses on there.*


With regards to pasture management, most participants’ responses alluded to good pasture management practices, with frequent faeces removal from pasture cited by most. Some cited rotational grazing, cross grazing with sheep, and harrowing. Poor field management was perceived to undermine and devalue the implementation of strategic controls, which in turn related back to cost. 


*So we have a barn of 20 horses and every single one of those 20 horses could go in the same paddock as mine, and they are all exposed to the same worms and eggs I suppose, so you know that’s a barrier for me, because what’s the point of me testing when the likelihood is that my horse is going to have a high burden because the paddocks aren’t poo picked, it’s going to waste more of my money.*
[HO1]

Examples of how to overcome barriers and facilitation of behaviour change were outlined.


*What I say at my yard is that we run a worm programme, they have to join. And if they don’t join it I want to know why, and if they don’t worm I want to know why. If they basically have a worm programme in place from a previous yard where they wormed and didn’t do egg counts, I persuade them to do the egg counts and to be honest, everyone comes over onto the egg counts. I have found a few barriers, people are a bit old fashioned aren’t they and they only like what they know, and they’re the type of people to just worm once a year and think that’s okay, but then actually when you do a worm count they’ve got a horribly high redworm count, or roundworm count and they’re like horrified.*
[YM6]

### 3.2. Theme 2: Helminth Information -and Anthelmintic Source

Two main subthemes were associated with helminth information and anthelmintic source: (1) point of sale and information, and (2) external influences.

#### 3.2.1. Subtheme 1: Point of Sale Information

This subtheme identified the information given at the point of anthelmintic sale. Many of the horse owners and yard managers advised that they did not receive any information at the point of sale. Some said the level of information depended on who was serving them.


*Depends who serves you. I’ve had information given in the past and sometimes I’ve not, I’ve just been handed it.*
[HO4]


*They don’t really give me a lot of information about to me I suppose because I think they must know that I know what I’m on about. To some people they might, but they don’t really tell me a lot about the products because I sort of know what I’m aiming at.*
[YM6]

For the few who had stated they received point of sale information, the information included:


*The dose, what wormer I should use, when I should give and she does tell me why as well, why I need to give it.*
[HO1]


*Just tells me which worms it kills. Pretty much it’s just sort of standard stuff isn’t it. It tells you up to 700 kg or and then just in theory what it kills.*
[YM1]


*They require you to give information about the horses, what size and age.*
[YM4]

#### 3.2.2. Subtheme 2: External Influences

This subtheme captures comments relating to the source of deworming information, deworming programme influences, and source of anthelmintics. Participants obtained information from variety of sources, including their veterinary surgeon, social media, and the internet; specifically, the British Horse Society and a customer facing laboratory were referenced. Many used a combination of the internet and their veterinary surgeon.

Often, the internet was cited as the source for general information, but if the participants had a specific concern, veterinary advice would be sought.


*If I was after general information, I’d probably just Google. If I thought they were severely affected by a worm burden I’d bring it up with a vet.*
[HO2]


*I’d google stuff like that usually and ask the vet if he was here at the time.*
[YM6]

In addition to a source of helminth information, veterinary surgeons were also cited by some as the main influence of their deworming programme:


*The vets, I always get veterinary advice on it.*
[YM4]

For some yard managers, they devised their yards’ helminth control programmes themselves. In one case, it was in conjunction with their veterinary surgeon.


*I guess it would be a collaborative effort, in conjunction with myself and the vets.*
[YM2]

For one horse owner, they have regular contact with an SQP positioned in online veterinary pharmacy, from whom they source both their information and anthelmintics.


*To be honest through I just speak to an SQP quite regularly to understand what I should do with him as I’m not very good with worming.*
[HO1]

Although HO1 used to shop around to find the best anthelmintic price, they now value the advice from the SQP with whom they have built a relationship and stick with the same source to purchase anthelmintics.


*I used to kind of shop around a bit, but the prices are very much similar really, it’s only a couple of pounds and you’re not buying them every week, it only every few months isn’t it so, but now I’ve tended to, I’ve found someone who gives more valid advice, and quite reliable and seems to know their stuff, I tend to stick with the one pharmacy that I get mine from.*
[HO1]

Conversely, some who sought veterinary advice then sourced their anthelmintics from an online veterinary pharmacy. Cost appeared to be a factor associated with choice of online retailer for some.


*Just get them from [Online veterinary pharmacy]. To be honest I just google cheap [wormers]. Depends which ones [the vets are] saying and I just google it and get from wherever is cheapest.*
[YM1]

For one horse owner, locality of the country store influenced their source of anthelmintics.


*I go the local suppliers, a suppliers for farmers, a local one.*
[HO3]

### 3.3. Theme 3: Accurate Dosing

This theme contained factors associated with accurate anthelmintic dosing; two main subthemes were identified: (1) problems associated with the administration of products, and (2) bodyweight.

Many referred to ensuring that hay was removed prior to product administration.


*I make sure he hasn’t got any food in his mouth first, so I pull any hay out of his mouth.*
[HO1]

The above was a recurring comment which many of the participants discussed.

Many also referred to ensuring they held the horse’s head up to encourage ingestion.


*I like to keep their head up until they have swallowed it.*
[YM4]

#### 3.3.1. Subtheme 1: Problems Administering

This subtheme describes challenges associated with anthelmintic administration. Many stated they did not have any problems, and some expressed confidence in their ability:


*I’m under five foot and I can worm any horse because I’ve got a technique.*
[YM3]

Others faced challenges due to horse behaviour:


*Well, it depends which one, there is a variety of different strategies. [One of the horses] won’t let you touch her nose with anything, so it has to be quite strategic. The rest of them, you can pretty much do headcollar-less. And sometimes try and give them a treat afterwards if I’m organised.*
[HO2]


*Multiple problems, especially with horses that aren’t keen with their mouths.*
[HO4]


*If it’s naughty I’d go to more efforts to control it, if we have any that are really naughty, we’d put it in their feed.*
[YM2]

Despite some horses presenting challenges, YM1 did not view them as a barrier:


*Yeah I’m pretty alright to be honest, I mean a few of them aren’t overly keen but its done before they know if you’re quick. So yeah I’m quite happy the way we do worm if we have to.*
[YM1]

Conversely, many expressed challenges:


*Er, either just wrestle them. Or try and put it in their feed maybe. But it’s a tricky one, especially if you’ve got a big horse that doesn’t want it being done. To be fair it’s not just the big ones, I’ve had a little tiny pony every year we have to worm put its legs over my shoulder, rearing up. It was a nightmare, I dread that time when I have to worm that pony.*
[HO4]


*You can’t wrestle with her, she’s too strong. We’ve got to touch it on her cheek, and scrub it, and then slide it down her face. And then like don’t stretch your hand out to push the wormer in yet because it’s just going to squirt all over the place because she’ll make you jump when she drags you along. And then just keep taking it away and back, so the whole thing takes about 10 minutes. I like push it on her cheek, take it away, push it a bit further down, take it away, and then in the end, insert it in her mouth a bit and shoot as fast I can and hold her head in the air really quickly afterwards.*
[HO2]

Specific concerns with regard to treating for *Anoplocephala spp*. were highlighted by one yard manager:


*I’ve got a couple that won’t have a syringe and I have to give granules. But you can’t with tapeworm, I struggle to tapeworm as there isn’t a granule for tapeworm.*
[YM6]

Two categories were associated with this subtheme: (1) underdosing, and (2) AR, though whilst HO3 considered it important not to underdose, they did not make the association to AR. This is likely explained by their lack of awareness of AR.


*What did someone say to me if they’re like 500 kg and you only give them enough for 400 kg then you needn’t have bothered giving them any at all as it’s not enough. You’ve got to give them the right amount for what they weigh, otherwise there’s no point in giving half of it. You’ve got to like give them the full amount that’s needed for the weight of the horse, I don’t know why but you’re supposed to do that.*
[HO3]

Others made the association between underdosing and AR.


*If you don’t administer enough wormer for the weight of your horse, then some worms can survive, hence leading to long term resistance.*
[HO1]

Some expressed concern over underdosing.


*I guess I don’t worry about overdosing, I do worry about underdosing. You’d rather get a bit more in because obviously there’ll be a bit on the face and a bit spat out or whatever.*
[HO2]


*I just happily over worm them all. As my vet has told me that it’s not a problem to over worm it’s a problem to under worm.*
[YM2]

Discussing how they ensure they do not underdose their horses:


*The only ones I don’t give a full wormer to are my little ponies, and I didn’t used to give a full wormer to my younger horses when they were smaller, so I guess I do adjust the dose to some extent but in terms of a 400 kg horse versus a 550 kg horse, I’d just give them both a full dose of wormer. My theory on that would be that they probably spit a bit out so they’d be better having the whole lot, rightly or wrongly.*
[YM2]

Interestingly, YM3 felt it was important not to overdose and associated this with financial effect:


*Yeah, it’s an important factor so we’re not giving a pony a full syringe but also for wastage, if we can get two ponies out of one syringe it’s going to cost a lot less.*
[YM3]

#### 3.3.2. Subtheme 2: Bodyweight

Bodyweight was considered by many as an important factor associated with deworming. Three main categories were identified within this subtheme, all associated with methods to identify bodyweight: (1) weight tape, (2) weigh bridge, and (3) visual estimate. Weigh bridges were a popular method to determine bodyweight amongst participants. A mixed approach to determining bodyweight was most discussed; however, one participant relied on a visual estimate only. Frequency of assessing bodyweight varied amongst participants; for some, frequent checks were made.


*I quite often frequent to the vets unfortunately for other instances, and every time I’m there I make sure he goes on the weighbridge. When my saddle fitter comes out he always uses a weigh tape every time so that’s another way. And often visually as well, because I have a bit of a baseline as to what he is, I can tell if he’s put weight on or if he’s lost weight.*
[HO1]


*We weight tape on a fairly regularly basis so we have a good idea. All our horses go on a weigh bridge twice a year.*
[YM4]

However, for others, it was less frequent or dependant on season:


*Not very often, only if we see any sort of problem. Obviously this time of year [late spring/summer] we’ve got so much grass at the minute so it’s important to keep an eye on the weight.*
[YM5]

The perceived accuracy of weight tapes, or lack of, was commented on, with one participant sceptical:


*The liveries use weight tapes, but I think they’re a bit of an average, I don’t think they give you a very accurate result.*
[YM6]

Yet another participant was confident with this method:


*I know what he weighs, I use my weigh tape.*
[HO3]

## 4. Discussion

This study, to the best of the authors’ knowledge, is the first to qualitatively explore horse owner and yard manager perception of the barriers affecting strategic helminth control. Results from this study include the opinions, attitudes, and beliefs of the participants, reflecting their experience and therefore providing invaluable and novel insight. In line with the qualitative research methodology, the results presented are not generalizable to the broader population. Through exploration and evaluation of experiences, as well as motivations for the determination of helminth controls, future initiatives could be shaped around motivational factors to facilitate positive changes to outdated practices. Moreover, to further understand stakeholder perception of the barriers affecting strategic helminth control, results from this study could be utilised by other researchers to create a meaningful survey-based approach for the purpose of building upon this initial exploratory work. Theme 1: Barriers to Adoption of Strategic Controls, identified a variety of complex factors, including personal resources, internal influences, and external influences perceived as barriers to preventing implementation of strategic controls by horse owners and yard managers. However, questioning additional perceived barriers on helminth information source and anthelmintic access, in addition to accurate dosing, were seen as key and important themes.

The Code of Practice for the welfare of horses, ponies, donkeys and their hybrids issued under section 15 of the Animal Welfare Act of 2006 (UK) indicates that parasite control programmes should be implemented by an SQP or veterinary surgeon and should include FWECs and appropriate anthelmintic use [39]. Additionally, both routine and indiscriminate use of anthelmintics are strongly discouraged [39]. The Code of Practice does not outline recommendations for saliva or blood testing to determine *Anoplocephala* spp. burden. Furthermore, “careful” pasture management, including rotational grazing and faeces collection, is considered an important component of an effective programme, in addition to quarantine measures for new horses [39]. In order to adopt a comprehensive strategic approach to the helminths of pathogenic concern, diagnostic methods—(1) FWECs, and (2) saliva tests—are required. However, despite the availability and ease of both tests, those that deemed that they operated a strategic control programme in this present study only carried out FWECs. Furthermore, instead of using saliva tests to determine *Anoplocephala* spp. burden, a blanket dose approach was utilised. It could be suggested that this is because there is not yet a reliable method to test for *Anoplocephala* spp. Resistance; however, this would seem unlikely, given that only two of those using FWECs also utilised reduction tests. The explanation for the lack of saliva testing uptake could be associated with several of the barriers identified by the present study, such as cost and convenience, but research to specifically identify the decision not to include saliva testing in existing FWEC programmes would be beneficial.

In the absence of a readily available praziquantel-only product on the UK market, the only options for targeting *Anoplocephala* spp. control are to use a combination product containing either ivermectin, moxidectin, or a double dose of pyrantel; in turn, this forces the potential unnecessary use of anthelmintic action on cyathostomins. It could be hypothesised that the lack of a praziquantel-only product is contributing to increased AR spread, but this remains unclarified. Whilst praziquantel-only products are available if sourced by veterinary surgeons under the Special Import Scheme, they are likely more expensive, and they will not be available to purchase with the same buying power. Cost was identified as a significant barrier to strategic controls within this present study. Research indicates that most horse owners purchase anthelmintics through an SQP [17], and off-licence praziquantel-only products would not be available via this popular retail route.

This study found that administration of anthelmintics presented in an oral paste can prove difficult, often depending on the horses’ temperament and owner ability. Problems administering anthelmintics could negatively influence dosing accuracy, a factor associated with AR. In the sheep industry, inaccurate dosing is a well-recognized factor associated with AR; farmers that perform no calibration of drench guns and solely rely on their accuracy are more likely to have AR on their farms [40]. One participant in the present study specifically expressed difficulty administering products to target *Anoplocephala* spp., given the lack of diverse pharmaceutical formulations available. This study identified that for horses to whom it is difficult to administer oral paste, in addition to methods of restraint and paste disguise, some participants utilised other pharmaceutical formulations, such as granules or a liquid solution to add to feed. However, products containing praziquantel are only available in an oral paste form. In a study evaluating the stress response induced in horses when administered anthelmintics in paste or tablet form, it was found that tablet administration induced less stress in comparison to paste administration [41]. Although the study was not designed to compare the palatability between each formulation, it did conclude that minimising stress in both the horse and owner by using the tablet formulation may increase compliance and subsequent anthelmintic efficacy [41]. One participant in the present study expressed that they “dreaded” “deworming” one of their ponies, thus, if alternate formulations were available to reduce the “dread” and stress for horse owners, it would be extremely beneficial. Understanding the options to ease anthelmintic administration for horse owners could prove to be a worthwhile endeavour.

Positively, the present study highlighted consumer confidence towards available anthelmintics with regard to their safety margins, as some participants discussed “overdosing” in order to ensure efficacy. Interestingly, one yard manager avoided overdosing, not for safety fears, but to maximise syringe use for more horses from a cost-effective perspective. Increased prescriber involvement in accurate dosing, such as that in the livestock industry, is required. Should prescribers take the end user’s skill and experiences into account when advising on effective dosing to counteract ineffective dosing, industry stakeholder discussion is required. Challenges surrounding responsible online anthelmintic supply may become evident with increased prescriber involvement in dosing advice.

Questions surrounding the quality and existence of point-of-sale information, particularly with regards to online retail, have been raised [42]. Horse owners who purchase anthelmintics online are less likely to value prescriber knowledge and advice [17]. Results from this study found that online anthelmintic purchase was popular, which is supported by previous research [17]. Whilst a specific relationship between point-of-sale information and anthelmintic source was not identified in this study, the information gained on point-of-sale information, or rather the lack thereof, is concerning. For those who reported some point-of-sale information, an inconsistent approach was commented on, even from the same retailer. Furthermore, this study found that some horse owners shop around online for the best anthelmintic price, and thus source products from multiple retailers. Therefore, it can be assumed that even if point-of-sale information is given, it will not be consistent, as there is not a standard framework of point-of-sale information and advice for prescribers to issue. A full review and critical appraisal of point-of-sale information is required, and a one ‘best practice’ guidance for all prescribers to adhere to would be beneficial. Furthermore, this study found that some participants struggled to find even one source of reliable information, thus emphasising the requirement for a single best practice approach to ensure ease of accessibility and consumer confidence.

Reassuringly, this study found that many participants advised they would contact their veterinary surgeon if they were particularly concerned about a helminth infection. However, Google was cited as a source for general information, indicating the need for accurate and consistent online messaging to be provided by prescribers and the veterinary pharmaceutical sector.

Exploratory research into horse owner decision-making and the influence of the horse–human relationship is becoming more commonplace. A strong owner–horse relationship has been identified, with most owners considering their horse a family member [43]. The total amount spent per annum per horse ranged from GBP 314 to GBP 14,240; a considerable difference which potentially reflects the amount owners can or are prepared to spend on their horses. In addition to time, welfare, and personal obligations, finance was identified as an important theme affecting the decisions which owners made around key events in their horses’ lives [43]. Recently, horse owners have been urged not to economise on essential horse care as the cost-of-living crisis unfolds [44]. Given that the cost associated with strategic helminth control was identified in this study as a barrier to adopting those measures, and that previous research has identified cost to be a considerable factor informing anthelmintic choice [20], further research is warranted to explore how to overcome this barrier. Cross-industry stakeholder collaboration may be required to ensure affordability of diagnostic testing services.

In line with previous findings [25], this study found livery yard managers to be influential in both a positive and negative way with regards to helminth control. Some yard managers implemented strategic controls in accordance with their own terms and beliefs. One yard manager overcame perceived barriers by some of their clients with a “traditional” approach and achieved behavioural change by showing the “problem” to their clients, utilising FWEC results. Putting the COM-B model into practice, YM6 showed their capability, provided the opportunity, and, subsequently, gained motivation through understanding. Tradition was cited by some as to why strategic controls were not implemented a “we’ve always done it this way” approach. Reluctance to deviate from outdated practices has been found in farmer-based social science studies [45]. However, the results from a recent study [25] investigating the impact COVID-19 had on horse management in livery yards in the UK provide some confidence in the notion that traditional practices and behaviours can change for the better when yard managers are faced with challenging circumstances and threats to horse welfare. Within the agricultural industry, farmer–farmer extension is successfully used to disseminate information and influence practices and behaviours through a “lead farmer” [46,47]. In an equine setting, yard managers could adopt the “lead farmer” role, and thus disseminate information and awareness of equine AR to influence and change traditional practices. However, the lead farmer’s familiarity with the practice is vital [46], and therefore, targeting yard managers and other figures who are influential on yards to adopt the best practices must first be achieved. Given that research has found that yard managers may have more frequent contact with veterinary surgeons [48], engagement opportunity may be increased in comparison to individual horse owners. Quantitative research would prove valuable for ascertaining uptake of interest and motivation of yard managers across the UK. There is also an opportunity for stakeholder collaboration to support future yard manager education and engagement.

The present study also identified hesitancy to implement strategic controls, specifically for new horses without any worming history. This supports previous research that identified confusion surrounding quarantine guidelines [23]. The nature of the horse industry sees regular movement of horses between both owners and yards; therefore, it would prove advantageous to redefine and disperse quarantine guidelines to help overcome this barrier.

Results from this study found a high rate of utilisation of weigh bridges to determine body weight; however, visual estimates and weight tapes were also commonly used methods. Previous research has found weight tapes to be inaccurate [49], and horse owners consistently underestimate the bodyweight of their horses [50,51,52,53], which could impact accurate anthelmintic dosing. Although all participants made a nod towards assessing bodyweight, there was no real indication that there was a particular effort made prior to anthelmintic administration. One participant specifically referenced monitoring bodyweight according to the season; however, seasonal bodyweight monitoring was not mentioned by other participants. Future research would be beneficial for ascertaining whether horse owners monitor their horses’ weights depending on the season, given that many horse owners still operate blanket seasonal anthelmintic control programmes, and research has found that the prevalence of equine obesity is higher at the end of summer in comparison to the end of winter [54].

## 5. Conclusions

Results from this study offer valuable insight into horse owner and yard manager perceptions of the barriers affecting strategic helminth controls. The complexities and inter-relationships between themes, sub-themes, and categories were detailed. Theme 1: Barriers to Adoption of Strategic Controls identified a variety of complex factors, including personal resources, internal influences, and external influences perceived as barriers to implementation of strategic controls by horse owners and yard managers. However, additional perceived barriers on helminth information source and anthelmintic access, in addition to accurate dosing, were also seen as key and important themes. Challenges associated with anthelmintic administration and bodyweight estimation suggest that inaccurate dosing is prevalent, posing a threat to the decreasing rate of AR. Encouragingly, as documented in the discussion, behavioural change is possible, and techniques can be implemented to influence and change outdated practices. Yard managers could prove particularly beneficial in the battle against AR if their influence on yard policy and practices is used positively. Further research is warranted to determine yard manager motivation to encourage engagement. Industry-wide implementation of agreed-upon guidelines could support prescriber decision making and encourage uptake of effective strategic helminth controls. The barriers identified in this study could help to influence guidance given to horse owners through a better understanding of their beliefs and behaviours.

## Figures and Tables

**Figure 1 vetsci-09-00560-f001:**
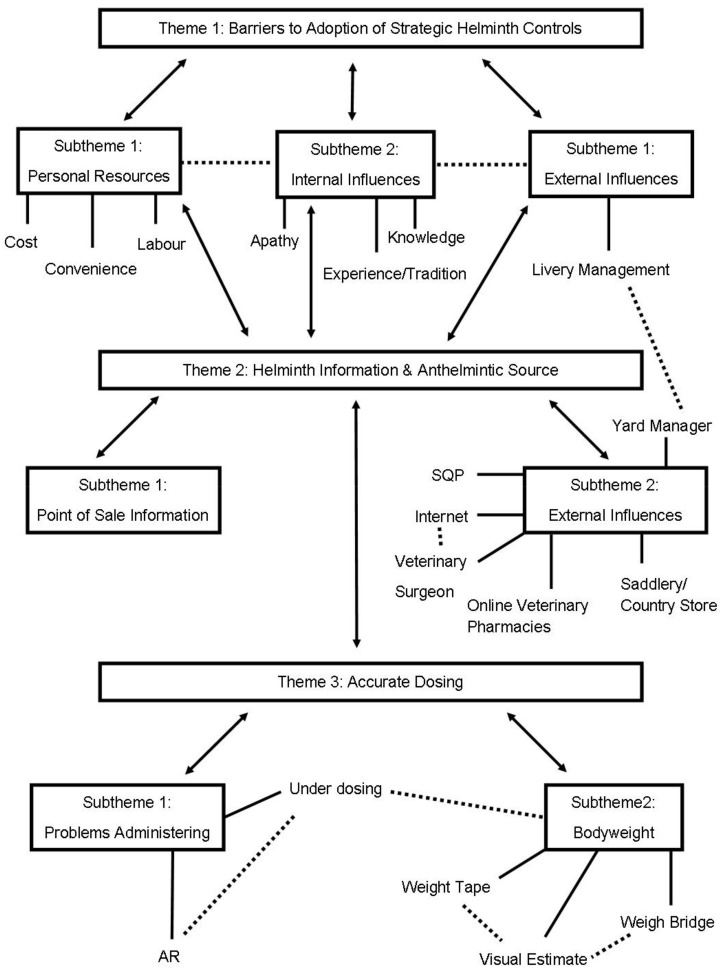
Thematic map illustrating the three main themes, subthemes, and categories, expressing their inter-relationships identified during thematic analysis of 11 semi-structured interviews.

**Table 1 vetsci-09-00560-t001:** Summary of the resistance status of the three authorised anthelmintic classes in the UK. [11,14].

Benzimidazoles	Macrocyclic Lactones	Tetrahydropyrimidines
Widespread cyathostomins resistance	Early indication of cyathostomins resistance	Reports of cyathostomins resistance
Early indication of *P. equorum* resistance	Widespread *P. equorum* resistance—particularly Ivermectin	Early indication of *P. equorum* resistance
	Early indication of *Oxyuris equi*	

**Table 2 vetsci-09-00560-t002:** Summary of participating horse owners (*n* = 5) and yard managers (*n* = 6) from England by length of time in the equine industry, number of horses, and livery management.

Unique Code	Participant	Length of Time in the Equine Industry	Number of Horses	Livery Management
HO1	Horse Owner	25 years	1	Full livery
HO2	Horse Owner	>25 years	5	DIY
HO3	Horse Owner	>22 years	2	DIY
HO4	Horse Owner	>10 years	1	DIY
HO5	Horse Owner	49 years	2	Full Livery
YM1	Yard Manager	>30 years	35	Retirement Livery
YM2	Yard Manager	>25 years	24	Full, Competition, Schooling, Rehabilitation, and Sales Livery
YM3	Yard Manager	>26 years	90	Full, Part, Assisted, and DIY Livery in addition to Riding School Horses
YM4	Yard Manager	>35 years	12	Full Livery
YM5	Yard Manager	>24 years	58	Full, Part, DIY, and Grass Livery
YM6	Yard Manager	>31 years	65	Full, Assisted, DIY, Holiday, and Schooling Livery

## Data Availability

Not applicable.

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
