# Peer review of "Equine Anthelmintic Resistance: Horse Owner and Yard Manager Perception of the Barriers Affecting Strategic Control Measures in England"

_vetsci, 2022, doi:10.3390/vetsci9100560_

Round 1

Reviewer 1 Report

The manuscript describes a survey carried out in UK throughout a semi-structured interviews aimed to the perceived barriers faced by horse owners and yard managers to adopting a strategic approach to anthelminthic control and explore their experiences. The identified barriers could be of help for the control of anthelmintic resistance and a more rational use of products effective against parasites. Furthermore, the results should be of help for a guidance to horse owners for the strategic control of parasite and the spreading of anthelmintic resistance. The study is original and deserves some interest for stakeholders involved in parasite control in horses. The discussion is quite long and boring, although it should be justified by the complexity of the results. In any case, my personal suggestion is to shorten it a bit to make more readable the manuscript.

I have no particular concerns, however the acronyms throughout the manuscript should be defined. In Table 2, I have counted 4 yards managers (YM-1…4) but in the table caption they are 6.

Reviewer 2 Report

General:

This is a unique paper that investigates an underserved area of equine management and delivery of sustainable health practices.  The authors demonstrate good knowledge of the inputs affecting the present anthelmintic resistance status of cyathostomin populations in managed horses and have correctly identified the determining agents for most health care decisions, i.e., horse owners and yard (stable) managers). 

The methodology of this study was intelligently structured, albeit probably under-powered given the relatively low number of subjects interviewed (5 owners and 6 managers). 

The structure of this paper is certainly unusual in the field of anthelmintic resistance and management recommendations.  For example, the information in Table 2 might be found more typically in the Results section of a paper.  Also, the Results present virtually no numerical data, but rather consist of an assortment of anecdotal responses to queries delivered during the survey.  As a parasitologist, this reviewer is more accustomed to seeing “hard data” in terms of frequency distributions, geometric mean population sizes, and significance of various statistical analyses.  The manner of presentation in this paper may well be very typical for research in the social sciences, but I am not familiar with those areas of endeavor.  I’m not saying the current format is correct or incorrect, just stating that it will seem quite unusual for scientists who study anthelmintic resistance or equine parasitology and might be reading this paper in future. 

Regardless, the authors present compelling information which highlights the depth and multiplicity of challenges to effecting sustainable parasite management in equine populations.  And rightly, these challenges are mostly centered in the behavior and thought processes of the relatively small number of humans who make decisions regarding the welfare of far greater numbers of horses. 

In addition, I believe the manuscript could be improved if the authors would consider the following recommendations and observations. 

Specific:

1.  The manuscript would benefit greatly from expert editorial attention.  For example:

a.    The paper currently contains numerous examples of subject/verb disagreement (e.g., line 62 “…is…no methods…”, lines 577-578 :Questions… has been raised…” 

b.    In many places, it seems that inclusion of a definite (the) or indefinite (a, an) article would have improved readability.  (Note: The reviewer is American, so this may just be due to colloquial differences.) 

c.     The use of semicolons in the text seems unusually frequent.  Two separate sentences might be preferable in some instances. 

Conclusion:

I believe this article is deserving of publication, but only the editors can decide whether Veterinary Sciences is the appropriate vehicle.  From a hard science standpoint, the low sample sizes for both populations interviewed are problematic.  I consider that the authors have developed a brilliant questionnaire and defined a methodology for subcategorization of those inputs.  In other words, they’ve done all the hard work of developing a tool and methods of analysis.  The next logical step would be to apply these methods to a far larger population of equine decision makers.  That would result in quantitative data regarding the various responses in several categories, and thereby help to prioritize the obstacles of human behavior that should be addressed.  This is an interesting and valuable report, but an optimal method of dissemination is for others to determine.  
